# Increasing the Sales of Suboptimal Foods with Sustainability and Authenticity Marketing Strategies

**DOI:** 10.3390/foods11213420

**Published:** 2022-10-28

**Authors:** Ilona E. De Hooge, Roxanne I. van Giesen, Koen A. H. Leijsten, Charlene S. van Herwaarden

**Affiliations:** 1Marketing and Consumer Behaviour Group (Bode 87), Wageningen University, P.O. Box 8130, 6700 EW Wageningen, The Netherlands; 2Consumer Research Unit, Policy Research & Analytics Department, Centerdata, P.O. Box 90153, 5000 LE Tilburg, The Netherlands; 3Marketing Group, Tilburg University, P.O. Box 90153, 5000 LE Tilburg, The Netherlands

**Keywords:** food waste, suboptimal products, sustainability positioning, authenticity positioning, marketing strategy, buying behaviour

## Abstract

To reduce food waste, it is essential to motivate consumers to purchase and consume products that deviate from optimality on the basis of only cosmetic specifications (also called suboptimal products). Previous research has shown it to be challenging to motivate consumers to buy such suboptimal products. Sustainability or authenticity positioning of suboptimal products may be a promising avenue, but no research to date has examined their effects on consumer *behaviour*. The current research examines whether sustainability and/or authenticity positioning increase the sales of suboptimal products and whether these strategies increase suboptimal product perceptions up to the level of optimal products. Two field experiments examined whether sustainability and authenticity positioning could increase the sales of suboptimal products in two settings: a daily market and a supermarket. They reveal that both types of positioning can increase the sales of suboptimal products. Moreover, in an online experiment, consumers were presented with suboptimal and optimal products with sustainability, authenticity, information, or no positioning, and consumers indicated their perceptions of and purchase intentions for suboptimal and optimal products. It demonstrates that the strategies motivate consumers to perceive suboptimal products as more similar to optimal products and can increase purchase intentions for suboptimal products. Together, these findings suggest that sustainability and authenticity positioning of suboptimal products can support the fight against food waste.

## 1. Introduction

It is gradually acknowledged that human natural resource usage exceeds planetary limits [1] and that reduction of resource usage is essential to fulfil the needs of current and future generations [2]. Food production is listed as one of the largest demanders of natural resources such as water, land, and energy [3,4]., and causes approximately one-third of all greenhouse gas emissions [5,6]. As about one-third to one-half of all produced food is wasted along the supply chain and in households [7,8], reducing food waste could easily reduce this resources usage [9]. Food waste can be defined as “…any food, and inedible parts of food, removed from the food supply chain to be recovered or disposed (including composted, crops ploughed in/not harvested, anaerobic digestion, bio-energy production, co-generation, incineration, disposal to sewer, landfill or discarded to sea)” [10]. Reducing food waste is stated as one of the necessary worldwide actions for a more sustainable future [2].

One of the essential causes of food waste is supply chain actors’ and consumers’ unwillingness to sell, buy, or consume suboptimal products. Suboptimal products, also called oddly-shaped or abnormally-shaped products [11,12], imperfect or imperfect-looking products [13,14] or ugly products [15,16], are products that diverge from perfect standards on the basis of their peripheral product aspects, such as appearance, but not on the basis of their main defining product aspects—intrinsic quality and safety [13,17]. Supply chain actors develop specifications that separate suboptimal products from perfect or optimal products [15,18]. These specifications categorise products into perfect/optimal and imperfect/suboptimal products on the basis of extrinsic product cues, such as shape or size [19,20]. Suboptimal products are subsequently mostly removed from the production line [18,19,21]. Supply chain actors’ motivation to apply cosmetic specifications is mostly based on the observation that consumers are unwilling to buy suboptimal products [22,23,24]. Therefore, motivating consumers to purchase suboptimal products would reduce food waste at all steps of the supply chain.

One promising avenue to increase consumer purchases of suboptimal products seems to be to apply a positioning strategy [25]. Recent research suggests that suboptimal products presented with a sustainability or authenticity positioning strategy may increase purchase intentions for suboptimal products [26]. Providing information on sustainability aspects related to food waste of suboptimal products or highlighting the product’s genuineness can increase purchase intentions. Yet, no research to date has examined whether such positioning strategies can affect consumer *behaviour* towards suboptimal foods. As the intention-behaviour gap exists for many (sustainable) behaviours [27,28], the examined purchase intentions may not translate into actual behaviour. The main purpose and novelty of the current research are, therefore, to examine whether a sustainability or authenticity positioning of suboptimal products can increase the sales of such products. We examine this research question with two field experiments: one experiment at a local market and one experiment at a supermarket.

Moreover, on a theoretical level, it is unclear how suboptimal products are presented with a sustainability or authenticity positioning compared to optimal products. No research thus far has examined whether it is possible to increase consumers’ quality perceptions of suboptimal products up to the level of optimal products with such a positioning strategy. The second goal and novelty of the current research are, therefore, to examine whether sustainability or authenticity positioning of suboptimal products can increase consumer perceptions of suboptimal products to the level of optimal products. We examine this with an online experiment in which consumers compare suboptimal with optimal products. Together, the findings of this multi-method approach provide valuable contributions to the current understanding of how sustainability and authenticity positioning strategies affect consumer responses to suboptimal products.

### 1.1. Suboptimal Products and Marketing Positioning Strategies

Suboptimal products are products that deviate from normal or optimal products on the basis of appearance standards (e.g., weight, shape, or size) [9,17,22], their date labelling (e.g., close to or beyond the best-before date) [17], or their packaging (e.g., a torn wrapper, a dented can) [14,15]. In all cases, the products deviate from peripheral aspects and not from intrinsic quality or safety [19,22,29]. The present research will focus on appearance deviations, such as bent cucumbers [17]. Consumers perceive suboptimal products to be of lower quality than optimal products, including being of lesser taste and of lower healthiness [12,15]. Consumers use aesthetic cues to make inferences about important attributes of products [12,30]. Aesthetic cues typically relate to the ‘what is beautiful is good’ notion [31,32], leading consumers to infer that suboptimal products are ‘not good’ and are, therefore, of lower quality compared to optimal products [12,33].

Contextual cues may mitigate the influence of aesthetics on quality perceptions of and on purchases of suboptimal products [12,15]. A contextual cue can be a marketing strategy with which the suboptimal products are positioned. Multiple scholars have recently provided suggestions on or examined whether presenting consumers with educational information on suboptimal products or positioning suboptimal products with contextual cues may have an effect on consumer responses to suboptimal products. For example, messages that boosted consumers’ self-confidence [34], or consumers’ feelings of guilt [24], messages focusing on communicating the products’ safety [11,14] or on moral consumption [35], have been found to have a positive effect on consumers’ purchase intentions for suboptimal products. Yet, these effects have not yet been well-confirmed [15]. One of the most promising ways to increase consumer perceptions of and purchase intentions for suboptimal products thus far seems to be to present suboptimal products with contextual cues concerning the sustainability issues surrounding these products [15,26,36,37]), or with contextual cues concerning the products’ authenticity or naturalness [15,26]). However, no research thus far has examined whether these positioning strategies actually affect consumer behaviour towards suboptimal products or whether they can motivate consumers to perceive suboptimal products as more similar to optimal products. Therefore, the current research focuses on whether sustainability positioning or authenticity positioning of suboptimal products can increase consumer purchases of suboptimal products and can increase the perceived similarity between suboptimal and optimal products.

When suboptimal products are presented with a sustainability positioning, purchase intentions for suboptimal products appear to be more positive compared to suboptimal products without a marketing strategy [26]. The sustainability positioning aims to make consumers aware of sustainability issues surrounding (the waste of) suboptimal products. In general, focusing consumer attention on a sustainability issue can increase awareness about the issue and motivate consumers to adjust their behaviour [7,38]. For example, sustainability information can reduce household food waste [39,40] or littering [41], increase willingness to pay for lower-carbon-footprint foods [42], and increase choices for environmentally friendly (sea)foods [41]. In the current case, highlighting sustainability aspects concerning suboptimal products can motivate consumers to increase purchase intentions for suboptimal products [26,36,37].

The second promising strategy that can increase purchase intentions for suboptimal products is an authenticity strategy [26]. Authenticity is a reference to what is genuine, real, and/or sincere [43,44,45]. Authenticity consists of multiple dimensions: authentic products are perceived as more natural or real, as locally or regionally made, as handmade or traditionally produced, and as reliable, sincere, and genuine [43,44,46,47,48]. For suboptimal products, especially, a focus on the naturalness dimension has positive effects on purchase intentions [26]. This dimension relates to the perceived healthiness and freshness of products and respect for the environment [49,50]. In general, natural products are perceived as more genuine, greener, and more organic [51], and authentic products are perceived as being of higher quality and freshness [52,53,54]. Therefore, an authenticity positioning increases both purchase intentions and also quality perceptions of suboptimal products [26].

### 1.2. The Current Research

Although the effects of sustainability and authenticity positioning on purchase intentions for suboptimal products seem promising, it is currently unclear whether such positioning strategies can also positively influence consumer behaviour towards suboptimal products. We, therefore, predicted the following:

**H1:** 
*A sustainability positioning of suboptimal products increases the sales of suboptimal products compared to no positioning.*


**H2:** 
*An authenticity positioning of suboptimal products increases the sales of suboptimal products compared to no positioning.*


We did not expect to find a difference between sustainability and authenticity positioning. To examine H1 and H2, we conducted two field experiments. In both field experiments, we presented suboptimal products with no positioning, sustainability positioning, or authenticity positioning and then kept track of the total amount of sales for both suboptimal and optimal products. Study 1 examined the sales at a local market, and Study 2 at a supermarket.

Suboptimal products only deviate from optimal products on the basis of peripheral product aspects, but consumers also perceive a quality difference between suboptimal and optimal products (De Hooge et al., 2017). Successful marketing strategies for suboptimal products should, therefore, also be able to decrease this perceived quality difference and motivate consumers to perceive suboptimal products as more similar to optimal products. Thus, we predicted:

**H3:** 
*Sustainability positioning decreases the difference between quality perceptions of suboptimal and optimal products compared to no positioning.*


**H4:** 
*Authenticity positioning decreases the difference between quality perceptions of suboptimal and optimal products compared to no positioning.*


As previous research has found authenticity positioning to have stronger influences on quality perceptions compared to sustainability positioning [26], we also predicted:

**H5:** 
*Authenticity positioning decreases the difference between quality perceptions of suboptimal and optimal products more than sustainability positioning.*


We examined H3–H5 with an online experiment (Study 3) in which consumers were presented with optimal and suboptimal products with either no, sustainability, or authenticity positioning. We also added an information positioning condition to examine whether providing any additional information on suboptimal products would already positively affect consumers’ quality perceptions of and purchase intentions for suboptimal products. We measured consumers’ quality perceptions of and purchase intentions for both suboptimal and optimal products and consumers’ perceived similarities between the two products. In all three studies, the suboptimal products deviated only in terms of shape (oddly shaped) from the optimal products (see Appendix A, Appendix B and Appendix C). Moreover, in all studies, we tested for significant differences between the different positioning strategies and between suboptimal and optimal products (either in terms of sales in Studies 1 and 2 or in terms of purchase intentions and quality perceptions in Study 3) using the 0.05 ratio.

## 2. Study 1: Sales of Suboptimal Products

### 2.1. Method

#### 2.1.1. Participants

Customers of a local market visiting a vegetable and fruit stand participated in the field experiment. They came to the market for their daily/weekly grocery shopping and were unaware of a study being run. We did not collect any individual information to avoid participant awareness. The market stand was located in the province of Gelderland (the Netherlands) and was present in the villages Herwijnen on Monday, Friday and Saturday (2500 inhabitants), in Vuren on Wednesday (1575 inhabitants), and in Leerdam on Thursday (19,253 inhabitants).

#### 2.1.2. Study Design

The field experiment took place in November 2019 (before COVID-19). The suboptimal products, pears, were provided by a local fruit and vegetable grower and supplied to the market stand every week. There were no suboptimal products sold at the market stand before our experiment took place. Customers visited the market stand and were presented with suboptimal and optimal pears, presented in separate crates in the market stand (see Appendix A). The suboptimal products deviated from the optimal products in terms of appearance (oddly shaped) and in terms of price (€1.50 per kilo for the optimal products, €1.00 per kilo for the suboptimal products). The suboptimal products were sold at a reduced price because the market stand owner only allowed us to run the experiment when a discount was provided for the suboptimal products.

Each week we manipulated positioning. A different positioning (Control, Sustainability, or Authenticity condition) was presented on a shelf display next to the suboptimal products. During the first week, the suboptimal products were presented without any positioning (Control condition). Week 2 presented the sustainability positioning: “Embrace imperfection: Join the fight against food waste!”. Week 3 presented the authenticity positioning “Naturally imperfect: Pears the way they actually look!”. These positionings were previously used in the study of Van Giesen & De Hooge [26], where the positionings were extensively tested.

#### 2.1.3. Purchase Measure

Customers’ purchases of the suboptimal and optimal products were measured as the amount of suboptimal and optimal pears sold in kilos per week. The staff of the market stand recorded the purchases of all products. To assess the relative increase in sales for suboptimal products, we subtracted the leftovers at the end of the week, damages, and decay from the original weight. We then compared the change in suboptimal product sales with the change in optimal product sales to assess the relative increase in suboptimal products.

### 2.2. Results

In total, 212 kilos of pears were sold during the experiment, of which 122 kilos were suboptimal products (57.5%). When no positioning was provided, 40 kilos of optimal products and 17 kilos of suboptimal products were sold (29.8% suboptimal products). The sustainability positioning decreased the optimal product sales to 22 kilos and increased the sales of the suboptimal products to 64 kilos (74.4% suboptimal products of total sales), reflecting an increase of 276% in sales of suboptimal products compared to no positioning.

The authenticity positioning accounted for a total sales of 28 kilos of optimal products and 41 kilos of suboptimal products (59.4% suboptimal products). Thus, both positioning strategies increased the sales of suboptimal products compared to no positioning (Overall effect: *Fisher’s Exact test p* = 0.001; sustainability vs. control: χ^2^ (1, *N* = 143) = 27.76, *p* < 0.001; authenticity vs. control: χ^2^ (1, *N* = 126) = 11.01, *p* = 0.001), supporting both H1 and H2 (the weight cases procedure to enter data in SPSS was used). The sustainability positioning also increased the sales of suboptimal products compared to the authenticity positioning (44.6 percentage point increase; sustainability vs authenticity: χ^2^ (1, *N* = 155) = 3.94, *p* = 0.05). 

When also reflecting on the changes in the sales of optimal products following a positioning, it is possible to make comparisons between the before-and-after changes in sales of (sub)optimal products and to estimate the overall impact of a positioning strategy. Following the sustainability positioning, there were 29 kilos more *suboptimal* products sold than optimal products, see Figure 1. Thus, there was both an increase in the suboptimal product sales (64–17 kilos), and a decrease in the optimal product sales (22–40 kilos), resulting in a total effect of 29 kilos of additional sales (47–18 kilos).

Following the authenticity positioning, there were 12 kilos more *suboptimal* products sold than optimal products. Again, there was a suboptimal product sales increase (from 41–17 kilos) and an optimal product sales decrease (from 28–40 kilos), revealing a total effect of 12 kilos of additional sales (see also Figure 1). In sum, both positioning strategies positively affected suboptimal product sales.

### 2.3. Discussion

Study 1 provides the first evidence that sustainability and authenticity positioning on the market can increase consumer purchases of suboptimal products, supporting H1 and H2. Study 2 was conducted to examine the effects in a different shopping setting and without a price discount. Moreover, it might be possible that the positioning strategies motivate customers to buy more (suboptimal) pears but that they negatively affect the sales of other products. To develop insights into this possibility, Study 2 compares the sales following the positioning strategies with the sales of previous years.

## 3. Study 2: Sales of Suboptimal Products at Supermarkets

Study 2 had multiple aims. First, Study 2 tested whether the positioning strategies increased suboptimal product purchases when no discount was provided (H1 and H2). Second, Study 2 aimed to generalise the findings of Study 1 to a different market setting. A local market may attract consumers that are more open to suboptimal products. Therefore, Study 2 focused on a local supermarket. Third, the positioning strategies in Study 1 were presented in a certain order, which may have affected the results. Therefore, Study 2 applied a different sequence of positioning strategies.

Finally, Study 2 aimed to investigate the effectiveness of the different authenticity dimensions. Authenticity encompasses various dimensions, including the product’s origin and naturalness [44,46]. Focusing on other dimensions of authenticity might be more effective [55,56]. Therefore, Study 2 investigated the effects of an authenticity positioning focusing on naturalness (similar to Study 1) and focusing on the local origin [57].

### 3.1. Method

#### 3.1.1. Participants

Customers of a local supermarket located in Helden (the Netherlands, 6265 inhabitants) visiting the vegetable and fruit section participated in the field experiment. They came to the supermarket for their daily/weekly grocery shopping and were unaware of a study being run. We did not collect any individual information.

#### 3.1.2. Study Design

The field experiment took place in November 2019 (before COVID-19). The suboptimal cucumbers were provided by Multigrow and supplied to the supermarket every week. The supermarket did not sell suboptimal products before our experiment took place. Visiting customers encountered a cucumber stand with two sections: one with optimal and one with suboptimal cucumbers (see Appendix B). The suboptimal products only deviated in terms of appearance (oddly shaped; both products €0.65 per cucumber).

To manipulate positioning, each week (from Monday to Friday), a different positioning was presented on a shelf display next to the suboptimal products. A smaller A5-format display closer to the suboptimal products also showed the positioning strategy. In week 1, the suboptimal cucumbers were presented without any positioning (control condition). In week 2, an authenticity positioning focusing on the local origin was provided: “Grown locally: grown with love, harvested with pleasure!”. In week 3, this concerned an authenticity positioning focusing on the naturalness of the products: “Naturally imperfect: Cucumbers the way they actually look!”. It is important to note that, during week 3, a previously scheduled cucumber promotion also took place. All cucumbers, both optimal and suboptimal ones, were on discount during this week (from €0,65 to €0,45). In week 4, the sustainability positioning “Embrace imperfection: Join the fight against food waste!” was provided.

The authenticity and sustainability positionings applied in the current field experiment were previously used in the study of Van Giesen & De Hooge [26]. Moreover, a pilot study run in the lab using the materials of Study 3 showed that also the positioning focusing on naturalness could increase purchase intentions for suboptimal cucumbers (M = 4.59) compared to the control condition (*M* = 4.17, *p* = 0.019; overall effect: *F* (4, 149) = 5.42, *p* < 0.001, *η*^2^ = 0.13), and could increase quality perceptions (*M* = 4.82, control: *M* = 3.33, *p* = 0.002; overall effect of positioning: *F* (4, 149) = 6.33, *p* < 0.001, *η*^2^ = 0.15). On Saturday and Sunday, the suboptimal products were removed to reduce the likelihood of habituation effects.

#### 3.1.3. Purchase Measure

Sales of suboptimal and optimal products were measured as the number of suboptimal and optimal cucumbers sold per week. The optimal products were registered in the supermarket’s ERP system. The suboptimal products could not be registered. Therefore, the number of suboptimal products remaining at the end of each day was subtracted from the inventory of suboptimal products at the beginning of each day while controlling for any losses due to damages or decay.

### 3.2. Results

#### 3.2.1. Sales

In total, 1232 cucumbers were sold during the experiment, of which 275 were suboptimal cucumbers (22.3%). A chi-square analysis of positioning on sales (suboptimal or optimal products) was conducted with the weight cases function to include the frequency data. Positioning influenced the sales of suboptimal products (χ^2^ (3, *N* = 1232) = 13.94, *p* < 0.01; standardized adjusted residuals: no positioning = −0.1, local origin = −0.9, naturalness = −2.2, sustainability = 3.6). Without a positioning strategy, 58 suboptimal and 204 optimal products were sold (22% suboptimal products, Table 1). The sustainability positioning increased the sales of suboptimal products most: it increased the sales of suboptimal products compared to no positioning (χ^2^ (1, *N* = 508) = 5.01, *p* = 0.03; 76 suboptimal (31%) and 170 optimal products), compared to the local origin positioning (21%, χ2 (1, *N* = 483) = 7.16, *p* < 0.01), and compared to the naturalness positioning (19%, χ2 (1, *N* = 733) = 12.82, *p* < 0.001). This result supported H1. The local-origin or naturalness authenticity positioning did not increase the sales compared to no positioning (χ^2^ (1, *N* = 499) = 0.26, *p* = 0.607, 20% suboptimal products; χ^2^ (1, *N* = 749) = 0.98, *p* = 0.323, 19% suboptimal products, respectively), failing to support H2. Similar results were found when taking into account the changes in sales of the optimal products. Together, these results suggest that especially sustainability positioning can positively influence consumer purchases of suboptimal products. 

#### 3.2.2. Additional Insights

As an indication of whether the suboptimal product sales replaced the sales of optimal products or were additional sales, we compared our sales data to the sales data of optimal cucumbers from exactly the same weeks in the previous year. This comparison suggested that customers purchased suboptimal products as additional purchases. The total sales during the same time period in 2018 amounted to 1003 cucumbers, compared to 1232 in 2019.

### 3.3. Discussion

Study 2 reveals that consumers can be willing to purchase suboptimal products and that especially a sustainability positioning increases suboptimal product sales (supporting H1). Unexpectedly, the two types of authenticity positioning did not have an effect (rejecting H2). For the local origin positioning, this may indicate that other aspects of authenticity might work better for suboptimal products. The absence of an effect of a positioning focusing on naturalness contradicts the findings of the previous study. The presence of a price discount promotion, which was held for all cucumbers (both suboptimal and optimal ones) during the week of the naturalness authenticity positioning, may have overshadowed the effects of this positioning. Future research is necessary to further examine the authenticity effects on suboptimal product sales. Study 3 was conducted to examine whether an effect of authenticity on consumer responses to suboptimal products can be found in a different setting. Moreover, one remaining question is whether the positioning strategies can increase consumers’ quality perceptions of and purchase intentions for suboptimal products up a level comparable to optimal products (H3–H5). Therefore, we ran Study 3.

## 4. Study 3

Study 3 investigated how consumers compare suboptimal products to optimal products when being presented with a positioning strategy. We measured consumers’ quality perceptions of and purchase intentions for both suboptimal and optimal products and asked consumers to compare suboptimal with optimal products. It is thereby possible that adding additional information to suboptimal products, independent of whether this information concerns sustainability or authenticity or any other information, may already increase quality perceptions of and purchase intentions for suboptimal products. Therefore, we included two control conditions: one condition without any information (similar to [26]) and one condition with information that could draw attention to suboptimal products.

### 4.1. Method

#### 4.1.1. Participants and Design

A total of 603 Dutch inhabitants (50.4% female, *M_age_* = 29.72, *SD_age_* = 10.47) participated in exchange for a small reward via the recruitment platform Prolific. They were randomly assigned to one of the conditions of a 4 (Positioning: Sustainability vs. Authenticity vs. Information vs. Control) × 2 (Product: Fruit vs Vegetable) between-subjects design with product choice, purchase intentions for both suboptimal and optimal products, quality perceptions of both suboptimal and optimal products, and similarity perceptions as the dependent variables.

#### 4.1.2. Procedure and Variables

The participants fulfilled an experiment similar to Van Giesen & De Hooge (2019), Study 2. They imagined doing their weekly grocery shopping at their local supermarket and buying apples or carrots. They encountered images of two shelves filled with apples (Apple condition) or carrots (Carrot condition): one shelf with optimal (perfect-looking) products and one shelf with suboptimal products (see Appendix C). The Sustainability condition read, “Apples [carrots] with special shapes: Don’t let them be wasted!” and the Authenticity condition read, “Directly from the tree (field): apples (carrots) with natural shapes!”. These slogans were identical to the ones used in Van Giesen & De Hooge (2019), Study 2. The Information condition read, “Craving for a snack? Take an apple (carrot)”. The Control condition did not show any text. A pilot study showed that suboptimal products presented with the information slogan were similarly rated in terms of taste (*M* = 6.29 vs. *M* = 6.26), health (*M* = 6.61 vs. *M* = 6.70), safety (*M* = 6.46 vs. *M* = 6.54), quality (*M* = 5.81 vs. *M* = 5.50), and expensiveness (*M* = 5.12 vs. *M* = 5.41) as suboptimal products presented without any text (all *t*s (228) < 0.96, all *p*s > 0.34). In the main experiment, the suboptimal products had the same price as the optimal products.

To measure *Product choice*, the participants indicated which product they chose to buy by clicking on the preferred category. To measure *Purchase intention for suboptimal products*, the participants saw only the suboptimal products and indicated how likely it was that they would buy those products (1 = not at all likely, 9 = very likely). They then indicated to what degree they thought that the product was of a very bad (1) or very good taste (9), very unhealthy (1) or very healthy (9), very unsafe (1) or very safe (9), of very bad quality (1) or very good quality (9), and to what degree they received value for their money when buying this product (1 = hardly any value; 9 = much value) (together averaged into one measure of Quality perceptions of suboptimal products, α = 0.90). Next, the participants saw pictures of the optimal products and answered the purchase intentions and quality perceptions of items for the optimal products. We calculated difference scores by subtracting the purchase intentions for and quality perceptions of suboptimal products from those of the optimal products. Finally, the participants provided Similarity perceptions of suboptimal products compared to optimal products by indicating to what degree they thought the two groups of products were similar, different (recoded), resembling, divergent (recoded), and of a similar quality (1 = not at all, 9 = very strongly, together averaged into one scale, α = 0.85).

### 4.2. Results and Discussion

#### 4.2.1. Product Choice

There were no effects of Product (apple/carrot) in any of the analyses (all Fs < 0.44, *p*s > 0.72). Therefore, we collapsed the data across product types. A chi-square analysis of Product choice showed that Positioning influenced the product choice (χ^2^ (3, *N* = 603) = 45.48, *p* < 0.001, Table 2). Both Sustainability and Authenticity positioning strategies increased suboptimal product choices compared to the two control conditions (χ^2^s(1) > 16.89, *p*s < 0.001). There were no other differences (χ^2^s < 1).

#### 4.2.2. Purchase Intentions

A one-way ANOVA with positioning as the independent variable and purchase intentions for suboptimal products as the dependent variable showed an effect of positioning (F(3) = 8.65, *p* < 0.001). Contrast analyses revealed that both Sustainability and Authenticity positioning increased purchase intentions compared to the two control conditions (*t*s > 2.74, *p*s < 0.01). There were no other differences (*t*s < 1.11, *p*s > 0.27).

An ANOVA on Purchase intentions for optimal products showed that the positioning did not affect purchase intentions for optimal products (*F*(3) = 1.25, *p* = 0.29). An ANOVA on the *difference* in purchase intentions revealed that positioning influenced the difference (*F*(3) = 7.20, *p* < 0.001). Both Sustainability and Authenticity positioning decreased the difference between suboptimal and optimal products compared to the two control conditions (*t*s > 2.38, *p*s < 0.02).

#### 4.2.3. Quality Perceptions

An ANOVA on quality perceptions of suboptimal products showed that positioning had an effect (F(3) = 22.62, *p* < 0.001). Authenticity positioning increased quality perceptions compared to the other three conditions (*t*s(599) > 2.52, *p*s < 0.01). Sustainability positioning increased quality perceptions compared to the two control conditions (*t*s(599) > 4.07, *p*s < 0.001). 

Surprisingly, an ANOVA on Quality perceptions of optimal products showed that the positioning also affected perceptions of optimal products (*F*(3) = 3.97, *p* < 0.01). Sustainability positioning increased quality perceptions of optimal products (*M* = 7.27) compared to the control conditions (*t*s(599) > 2.46, *p*s < 0.02). An ANOVA on the *difference* in quality perceptions revealed that positioning influenced the difference (*F*(3) = 13.65, *p* < 0.001). Authenticity positioning decreased the difference compared to the other three conditions (*t*s(599) > 3.75, *p*s < 0.001), supporting H4 and H5. The difference found for authenticity positioning was not different from zero (*t*(152) = 1.36, *p* = 0.18). Sustainability positioning reduced the difference only compared to Information positioning (*t*(599) = 2.21, *p* = 0.03), partially supporting H3.

#### 4.2.4. Similarity Perceptions

An ANOVA on similarity perceptions revealed an effect of Positioning (*F*(3) = 7.61, *p* < 0.001). Both Sustainability and Authenticity positioning motivated consumers to perceive suboptimal and optimal products as more similar compared to no positioning and compared to Information positioning (*t*s > 2.81, *p*s < 0.01), supporting H3 and H4. There were no other differences (*t*s < 1).

### 4.3. Discussion

Study 3 extends the findings of Studies 1 and 2. Both sustainability and authenticity strategies increase consumers’ quality perceptions of and purchase intentions for suboptimal products, supporting H3 and H4. An authenticity positioning even increased quality perceptions of suboptimal products up to the level of optimal products, supporting H5. Finally, Study 3 shows that these effects cannot be attributed to ‘simply’ providing information.

## 5. General Discussion

Suboptimal products are frequently disposed of throughout the supply chain or sold at lower prices. These practices result in the devaluation of foods and food waste. More sustainable solutions focus on ways to increase consumer acceptance of suboptimal products, thereby reducing both supply chain actors’ and consumers’ food waste. The present research extends existing knowledge on these solutions by demonstrating that sustainability and authenticity positioning can increase the sales of suboptimal products. Moreover, we demonstrate that especially an authenticity positioning increases consumers’ quality perceptions of suboptimal products up to a level where consumers hardly see a difference between suboptimal and optimal products.

### 5.1. Implications

Our research is one of the first to compare consumer responses to suboptimal products with those towards optimal products. Previous research mostly focused on consumers’ negative responses to suboptimal products and on how to motivate consumers to respond more positively [24,25,34,58]. Even though this research has provided many valuable insights, it has been unclear whether marketing strategies motivate consumers to be more positive about suboptimal products or to be less positive about optimal products. It has also been unclear whether marketing strategies can motivate consumers to perceive suboptimal products as similar to optimal products. The present findings reveal that marketing strategies for suboptimal products hardly affect consumer perceptions of optimal products. This insight is valuable for supply chain actors who are hesitant to apply marketing strategies out of fear of damaging the sales of optimal products [19]. Moreover, especially an authenticity positioning can motivate consumers to perceive suboptimal products as similar to optimal products. Thus, it might be possible to bridge the gap between suboptimal and optimal products. 

Our findings also reveal that the effects of sustainability and authenticity positioning strategies are not based on an information or attention effect. Even though attracting attention to regular products may positively affect consumers’ purchase intentions [59,60,61], attracting attention to suboptimal products by simply providing additional information is not enough to motivate consumers to purchase such products. Aesthetic deviations already attract consumer attention in a negative way [62]. The present findings reveal that sustainability and authenticity positioning are able to ‘counteract’ these negative attention effects and motivate consumers to purchase suboptimal products.

One of the essential steps in suboptimal product research is a move towards consumer *behaviour*. In the last couple of years, multiple studies have examined how consumer responses to suboptimal products can be changed using hypothetical scenarios. Even though these findings have provided valuable first insights, it is essential to examine to what degree such findings also translate to consumer behaviour. Indeed, multiple scholars have mentioned the value of suboptimal products’ sales data and raised the importance of generating such data preferably across different sales channels [15,19]. Our findings reveal that such data may generate novel insights compared to purchase intention data and may be a promising avenue for future research.

### 5.2. Limitations and Future Research

While we have demonstrated the effectiveness of sustainability and authenticity positioning of suboptimal products on actual purchase behaviour, our studies also contain some weaknesses. The repeated measures design of our field studies, combined with the presence of a price discount in Study 1 and a general price promotion during the naturalness authenticity positioning in Study 2, limit the generalisability of our findings. In Study 1, the price discount was constant across experimental conditions and, therefore, can not explain the differences between the positioning strategies. At the same time, the findings of Study 2 may suggest that presenting multiple types of marketing strategies, such as a positioning strategy combined with a price discount, may limit the effectiveness of one or both strategies. It would thus be valuable to study the effects of positioning strategies combined with other marketing strategies in different market settings.

Moreover, the field studies currently do not provide some additional valuable insights into the effects of positioning strategies at different times and on other products. Our studies were mainly conducted on weekdays, not providing information on the sales for weekends. It may be possible that weekday grocery shoppers are different from weekend grocery shoppers or that positioning strategies draw more attention on the usually less busy weekdays compared to weekends. Also, even though the sales of Study 2 suggest otherwise, it may be possible that positioning strategies for one type of suboptimal product negatively affect the sales of other types of products. Future research is necessary to explore these possibilities.

Also, Study 3 used a hypothetical scenario design, which limits the generalisability of the findings to actual purchase behaviour. Indeed, whereas the field studies revealed the sustainability strategy to be the most effective, the lab study revealed the authenticity strategy to be the most effective strategy. It may be possible that consumers use different decision processes when making decisions in hypothetical scenarios compared to actual settings (e.g., more rational thinking processes versus more intuitive thinking processes) or that consumers are not very well able to reflect upon how they would behave in actual situations. Future research is necessary to examine the degree to which purchase intentions for suboptimal products, as demonstrated in the current and in previous studies, reflect actual consumer behaviour towards suboptimal products.

Future research should also investigate the effectiveness of our positioning strategies over time and in various offering settings. In our studies, suboptimal and optimal products were always offered on separate shelves, thereby highlighting the differences between suboptimal and optimal products. It might be possible that the positioning strategies work differently when both products are combined on one shelf or when consumers have repeatedly encountered these positioning strategies.

## 6. Conclusions

In sum, the present research has shown that it is possible to increase the sales of suboptimal products and to motivate consumers to view suboptimal products as more similar to optimal products with marketing positioning strategies. With these findings, our research provides the next step towards a better future for suboptimal products: a future in which suboptimality may no longer fall below optimality.

## Figures and Tables

**Figure 1 foods-11-03420-f001:**
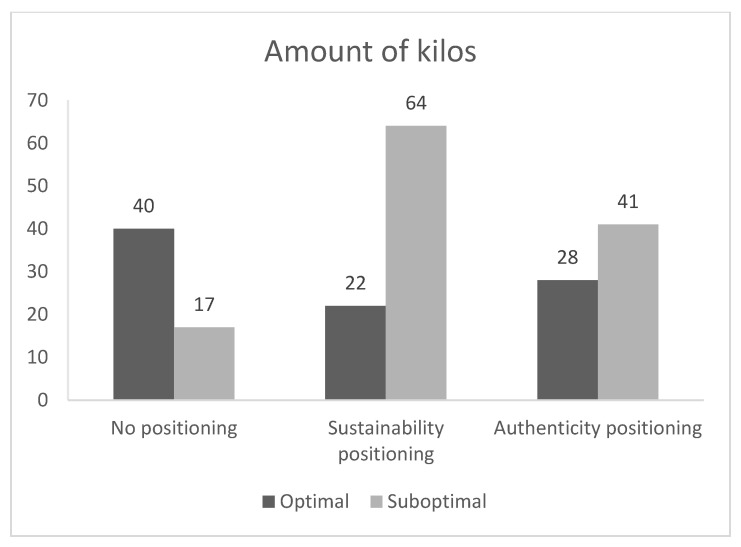
Suboptimal and Optimal Product Sales (in kg) in Study 1.

**Table 1 foods-11-03420-t001:** Suboptimal and Optimal Product Sales (in kg) as a function of Positioning Strategy in Study 2.

	Positioning Condition			
Dependent Variable	No	Local Origin	Naturalness	Sustainability
	(Control)	Authenticity	Authenticity	
	Week 1	Week 2	Week 3	Week 4
	% (n)	% (n)	% (n)	% (n)
Sales suboptimal products	22% (58) ^a^	20% (48) ^a^	19% (92) ^a^	31% (76) ^b^
Sales optimal products	78% (204)	80% (189)	81% (394)	69% (170)
Ratio suboptimal/optimal sales	0.28	0.25	0.23	0.45
Difference suboptimal sales–no positioning		−2% (−3 pts)	−3% (−5 pts)	+9% (+17 pts)

The ratio in suboptimal sales was calculated by dividing the suboptimal product sales by the optimal product sales for each of the positioning strategies. Different superscripts (e.g., a and b) indicate a statistically significant difference.

**Table 2 foods-11-03420-t002:** Product choice, Purchase intentions, Quality perceptions, and Similarity perceptions Means (and SDs) for (Sub)optimal products as a function of Positioning strategy in Study 3.

	Positioning Condition			
Dependent Variable	No (Control) *n* = 152	Information (Control) *n* = 149	Sustainability *n* = 149	Authenticity *n* = 153
Product choice suboptimal products	% (*n*) 9% (14) ^a^	% (*n*) 7% (10) ^a^	% (*n*) 28% (41) ^b^	% (*n*) 31% (47) ^b^
Purchase intention suboptimal products	*M (SD)* 4.40 (2.27) ^a^	*M (SD)* 4.30 (2.11) ^a^	*M (SD)* 5.11 (2.20) ^b^	*M (SD)* 5.39 (2.36) ^b^
Purchase intention optimal products	7.51 (1.47) ^a^	7.35 (1.39) ^a^	7.35 (1.48) ^a^	7.18 (1.50) ^a^
Difference optimal–suboptimal Purchase intentions	3.11 (2.98) ^a^	3.05 (2.71) ^a^	2.24 (2.91) ^b^	1.79 (3.15) ^b^
Quality perceptions suboptimal products	5.72 (1.50) ^a^	5.65 (1.59) ^a^	6.42 (1.47) ^b^	6.86 (1.44) ^c^
Quality perceptions optimal products	6.78 (1.34) ^a^	6.89 (1.36) ^a^	7.26 (1.14) ^b^	7.03 (1.30) ^ab^
Difference optimal–suboptimal Quality perceptions	1.06 (1.61) ^a^	1.24 (1.76) ^a^	0.84 (1.31) ^b^	0.17 (1.54) ^c^
Similarity perceptions	4.10 (1.49) ^a^	4.21 (1.48) ^a^	4.78 (1.56) ^b^	4.70 (1.64) ^b^

Product choice reflects the percentage of respondents who chose the suboptimal product above the optimal product. Purchase intentions means (and sd) reflect the respondents’ likelihood of buying the (sub)optimal product (1, not at all likely, 9, very likely), quality perceptions reflect respondents’ perceptions of the (sub)optimal product (1–9), and similarity perceptions reflect respondents’ perceptions of the suboptimal product being similar to the optimal product (1, not at all, 9, very strongly). There are no significant differences between means with the same superscript, with χ^2^s and *t*s < 0.64, *p* > 0.42. Means with different superscripts differ significantly with χ^2^s > 16.89, *p*s < 0.001, and *t*s > 2.53, *p*s < 0.02.

## Data Availability

Data is contained within the article, and can be made available upon request.

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
