# Peer review of "Increasing the Sales of Suboptimal Foods with Sustainability and Authenticity Marketing Strategies"

_foods, 2022, doi:10.3390/foods11213420_

Round 1

Reviewer 1 Report

The paper postulate that one way to reduce food waste is to increase the sales of suboptimal food products, foods with aesthetic characteristics. Consumers perceive these products had a lower quality than optimal ones. The authors test two ways to stimulate the sales of suboptimal food products thought sustainability and authenticity marketing arguments. They use three studies, two with real products based on sales data (without individual data) and the last based on declarative online data.

The whole paper is interesting but a detailed reading is necessary to correct and precise some elements.

I do not have specific comment about the Introduction. It is interesting, even if it could be more precise.

Study 1. The description is clear. The product is optimal and suboptimal pears on a stand of a local market (in the Netherlands). The experiment took place in November (pears are in season) during three weeks: (1) control, (2) sustainability positioning, and (3) authenticity positioning. Whatever is the treatment, the prices remain the same: 1.5€/kg for optimal pears and 1.0€/kg for suboptimal pears.

- We understand that before the experiment, the seller did not have suboptimal products. Is it right? It is necessary to well explain the modification generated by the experiment in the natural context.

- It could be informative to know some additional elements about the market: the place (height of the town), the frequency of the market (daily, weekly), etc.? In order to well understand the context.

- It would be more informative to present the results with only one figure (rather than 2) and histograms in order to better understand these results.

- Authors could indicate the full sales: 57 kilos of pears in control, 86 kilos in sustainability (+50.8%), and 69 kilos in authenticity (+21.1%).

- For sustainability, the total effect is not “65 kilos of additional sales” (p.5, l.227), but 29 kilos: (22 – 40) + (64 – 17) = -18 + 47 = 29.

- For authenticity, the total effect is not “36 kilos of additional sales” (p.6, l.237), but 12 kilos: (28 – 40) + (41 – 17) = -12 + 24 = 12.

- Do you have any information on the sales of other fruits with this seller. Customers buy more (suboptimal) pears, but do they buy less other products? Could you discuss this potential side effect?

Study 2. The description is relatively clear. The product is optimal and suboptimal cucumber in a supermarket (in the Netherlands). The experiment took place in November (not sure that cucumbers are in season) during for weeks: (1) control, (2) authenticity local, (3) authenticity natural, and (4) sustainability.

- It could be informative to know some additional elements about the supermarket: the place (height of the town), profile of customers, etc.? In order to well understand the context.

- Did the author have information about the volume of sales on Saturday and Sunday? (p.7, l.285-286) It could be interesting for the analyze. The buyers are not the same on week and on weekend.

- The price of the cucumber is not very clear. The authors underline that the price is the same for both products: 0.65€/cucumber (p.7, l.276). But they write (p.8, l.322-323): “the suboptimal products were sold at a reduced price”. The interpretation of results is not the same.

- The Table I contains many information, and some of them are not really informative. It will be better to give the ratio “suboptimal sales / optimal sales” rather than difference, because volumes are not comparable from one week to other. The argument is the same for the two last lines. What is the information? You could write (for instance):

----------------------------------- No   ---  Local --- Nature --- Sustainability

Ratio subopt/opt                --- 0.28 --- 0.25    --- 0.23     --- 0.44

Diff subop sales – no pos. ---         --- -2pts   --- -3pts    --- + 9pts -points of percentages)

- Could the authors give more information about the promotion on week 3? What is this promotion (price, quantity), and is it the same on optimal and suboptimal products?

Study. The description is relatively clear. It is an online questionnaire about optimal and suboptimal apples and carrots. The two products are presented to all participants in four treatments: (1) control, (2) neutral information, (3) sustainability, and (4) authenticity.

- the explanation of questions is not very clear. The authors write (p.9, l.389-390): “to measure purchase intention, the participants saw only the suboptimal products and indicates how likely it was that they would by those products” (I understand they only measure the PI for suboptimal product), and after (p.9, l.397..): “we calculate differences score by subtracting the purchase intention [..] for suboptimal products from those for the optimal products” (I understand that the authors have the two measures).

- What are: “Quality perception, alpha=.90” (p.9, l.395) and “alpha =.85” (p.10, l.402)?

- Table II is not very clear about results; I think that graph would be more informative. It is important to provide information about optimal purchase intent and optimal quality perception, in order to well understand the analyzes. For optimal product, whatever the treatment, the PI is relatively constant. Histograms would be more informative about differences observed.

- I do not understand a part of footnote under Table II (p.10, l.433-438).

- What are the limits of Study 3 (in section 6, page 12)?

Last, it could interesting to underline and discuss that if in hypothetical context (study 3) authenticity works better than sustainability, in real context this is the reverse.

Minor comments

- p.1, l.4: what is the footnote “a” for the first author?

- p.5, l.204 : it is not “2.2. Results and discussion” but “2.2. Results”. Because section 2.3 presents the discussion.

- p.7, l.298 : it is not “3.2. Results and discussion” but “3.2. Results”. Because section 3.3 presents the discussion.

- p.8, l.313: where is the footnote 3?

- p.8, l.317: are the sales of cucumber in kg or in number? (I understand that it is in number).

- p. 8, Table I: could the author give information about “a” and “b”? (level of test, I suppose). The same for Table II.

- p.10, l.405: the authors could write “there were no effect of product (apple / carrot) in any…” in order to recall the products to the readers.

Reviewer 2 Report

I very much enjoyed reading this manuscript. I think the topic is highly relevant for this journal and of great importance in general. The authors have used research methods in a creative way. I only have a few comments.

1. I was not worried about the price difference in Study 1. First, in reality, consumers expect to pay less for suboptimal products. Second, this was constant across experimental conditions and therefore did not create any confounding.

2. We do not know if the sentences that were displayed were tested prior to conducting the studies. Obviously, they appear to be good manipulations, but checking this with a sample of consumers (or judges) would have been preferable.

3. In Study 3, I presume that images of apples and carrots were shown to the participants, but this is not mentioned clearly in the manuscript. If this was the case, it would be interesting to show these images in an appendix (taking pictures of the crates in Study 1 and Study 2 would have been relevant too).

4. I find the qualitative data of no interest. It is very anecdotical. We do not know how many consumers were interviewed, the questions they were asked, the conditions in which they were, etc. This does not bring anything of value to this otherwise very interesting research. 

In general then, a very good manuscript. Good luck with your future research!

Reviewer 3 Report

This study has practical relevance. Please find my specific comments

Abstract

What do you mean by Suboptimal Foods? Could you please add one sentence about it in the abstract

It looks general. Please add the significant findings. What are all the criteria/parameters followed in this survey?

What is the procedure followed in online experiment? Could you please briefly explain in the abstract

Keywords: Avoid the words used in the title

Introduction

Update the old references (published before 2018) with recent reference

Summarize the findings of similar studies and add the novelty of this study

Study design

Whether the study was included the opinon/interest of different age groups of people on Suboptimal Foods

What are all the major parameters/criteria were considered during this study?

Any statistical analysis was conducted to understand the significant between the consumers preference?

Results and discussion

Authors have highlighted only about the results. Please improve the discussion part

Reference

References need to be updated

Round 2

Reviewer 1 Report

Dear Authors,

Thank you very much for your reply. The new version answer to my asks.

I will be clearer about table I (pages 9-10).

The "difference suboptimal sales - no positionning" is not on "ratio" line, but on the "sales suboptimal products" line (pts are points of percentage)

-2pts = 20% - 22%

-3pts = 19% - 22%

+9pts = 31% - 22%

Author Response

Dear Reviewer 1,

thank you for your kind words, and for further clarifying your comments about the numbers in Table 1. Now we understand what you meant, our apologies for this confusion. We have now added the % differences to Table I. Thank you! 

Reviewer 3 Report

The authors have taken a great effort to revise the manuscript. I really appriciate the authors effort. The revised version is suitable for publication. 

Author Response

Dear Reviewer 3,

thank you very much for your positive words. We are happy that our revision met your expectations. Thank you once again for your very useful comments: they have substantially improved the manuscript!